# The Importance of Water for Purification of Longer Carbon Nanotubes for Nanocomposite Applications

**Vladimir Z. Mordkovich *** , **Maxim A. Khaskov** , **Veronika A. Naumova, Victor V. De, Boris A. Kulnitskiy and Aida R. Karaeva**

Technological Institute for Superhard and Novel Carbon Materials, 7a Tsentralnaya Street, Troitsk, 108840 Moscow, Russia
* Correspondence: mordkovich@tisnum.ru

**Abstract:** Ultralong carbon nanotubes (UCNTs) are in high demand for nanocomposites applications due to their magnificent physical and chemical properties. UCNTs are synthesized by the catalytic chemical vapor deposition (CCVD) method and, before use as fillers in nanocomposites, should be purified of residual catalyst and non-CNT particles without significant destruction or scissoring of the UCNT. This study investigates the role of water vapor for purification of UCNTs from iron catalyst particles and the importance of water assistance in this process is confirmed. It was shown that wet air treatment of products of UCNTs CCVD synthesis under mild conditions can be used to sufficiently decrease residual iron catalyst content without significant carbon losses in comparison to the results obtained with dry air, while the residual iron content was shown to significantly influence the subsequent oxidation of different forms of carbons, including UCNTs. The increasing of D/G ratio of Raman spectra after wet air treatment of products of UCNTs CCVD synthesis makes it possible to conclude that iron catalyst particles transform into iron oxides and hydroxides that caused inner structural strains and destruction of carbon shells, improving removal of the catalyst particles by subsequent acid treatment. UCNTs purification with water assistance can be used to develop economically and ecologically friendly methods for obtaining fillers for nanocomposites of different applications.

**Keywords:** nanocomposites; ultralong carbon nanotubes; catalytic chemical vapor deposition; purification; catalyst removal; water vapor treatment; thermal analysis; Raman spectroscopy

## 1. Introduction

The outstanding physical and chemical properties of carbon nanotubes (CNTs) open prospects for their use in different nanocomposites for various industrial applications such as biosensors, supercapacitors, solar cells, EMC shielding, corrosion protection materials, etc. [1]. On the other hand, some properties of nanocomposites, for example mechanical or electrical ones, are highly dependable on the aspect ratio of CNTs and the longer the nanotubes, the better the properties of nanocomposites [2,3]. As such, ultralong CNTs are in urgent demand for many responsible applications. There are two main methods of synthesis of ultralong CNTs. One is carried out on flat substrate [4,5] and the other is performed in a fluidized-bed reactor with floating catalyst and aerogel [6]. Unfortunately, since both methods are based on catalytic chemical vapor deposition (CCVD), non-CNT and residual iron catalyst particles [7] are formed in synthesis products besides ultralong CNTs. It should be noted that the purity of CNTs or the low content of non-CNT and catalyst particles often represent crucial demand for better properties of nanocomposites. For example, mechanical properties, such as Young's Modulus and toughness, are increased in polymer nanocomposites when purified CNTs are used when compared with raw CNTs [8]. The purity of carbon nanotubes also has a significant influence on the transport properties of CNT-based nanocomposites. For example, purification of CNTs can significantly improve thermal conductivity of epoxy-based polymer nanocomposites with CNTs [9].

There are many purification methods of CNTs, including chemical methods, physical methods and a combination of both [10]. Among them, gas phase oxidative purification by air is the most popular method used in the community for amorphous and non-CNT carbon removal due to its convenience and for economic and ecological reasons. Moreover, according to study [11], gas phase oxidation is necessary before any other investigated chemical purification methods as it causes oxidation of carbon shells, which envelop catalyst particles. The gas phase oxidation purification method of CNTs synthesis products is based on the fact that non-CNTs usually have higher oxidation activity in comparison with CNTs [12]. However, some metals, like iron or its compounds, which are used as catalysts for CNT synthesis, being at the same time the catalyst of carbon oxidation by oxygen and other oxidizing gases [3,8] can level out these differences between oxidation activity of CNTs and non-CNTs, resulting in very low yield of pure carbon nanotubes. To minimize the undesired oxidation of CNTs and increase the yield of final product before air oxidation, it is necessary to remove the residual catalyst particles [8]. According to study [11], SWCNTs can be purified from iron catalyst particles by wet air treatment under mild conditions, excluding carbon oxidation (at 225–425 °C), followed by acid treatment. The mechanism of this purification is based on density changes of iron under oxidation, which is 7.86 and 5.13 $cm^3$/g for metallic iron and iron (III) oxide, respectively. The decreasing of iron density resulting from oxidation causes swelling of encapsulated iron, destruction of the carbon shell and the exposure of iron for the subsequent acid treatment. It should be noted that this method was introduced only for purification of SWCNTs [13] and was only proven for use for SWCNTs by other researchers [8]. There are no publications on the purification of ultralong multiwall CNTs (UCNTs) by this method, despite the fact that UCNTs are quite appropriate for using in nanocomposites [14,15].

This work is devoted to study of water influence on purification of ultralong CNTs, obtained by water-assisted CVD method [7].

## 2. Materials and Methods

Carbon nanotubes (CNTs) were obtained by the water-assisted chemical vapor deposition method using ethanol ($C_2H_5OH$) as a carbon source, thiophene ($C_4H_4S$) as a growth activator, and ferrocene (($C_5H_5$)$_2$Fe) as a catalyst precursor [7]. The synthesis of the CNTs was carried out in a quartz reactor at a temperature of 1150–1200 °C in a stream of hydrogen, which was fed into reactor at a rate of 1.0 $m^3$/h, at a pressure of 1 bar. The three-component mixture ($C_2H_5OH$ + 0.5% $C_4H_4S$ + 1% ($C_5H_5$)$_2$Fe) was fed into the reactor using an ultrasonic nebulizer. This device converted the liquid reaction mixture into a fine aerosol that then entered the hydrogen stream (carrier gas) directed into the CVD synthesis reactor.

The mild oxidation of CNT synthesis products was carried out in a quartz tube reactor at 400 °C for 6 or 12 h under dynamic (50 mL/min) synthetic air atmosphere. Before dynamic synthetic air was inserted into reactor with CNTs, it was bubbled through distilled water at 50 or 80 °C. After mild oxidation, the samples were treated with 18 wt.% hydrochloric acid at 70 °C for 4 h (2 times for 2 h), rinsed with distilled water after acid treatment until pH of water was higher than 4 and then dried at 110 °C for 1 h. The heat treatment was carried out at muffle furnace under static air atmosphere at 480 °C for 2 or 6 h.

Thermogravimetric analysis was carried out using a NETZSCH STA 449 F1 device in an alumina crucible with the heating rate of 10 K/min under dynamic synthetic air atmosphere (50 mL/min). Raman spectroscopy was carried out using a Renishaw inVia Raman Microscope device (Renishaw, Warton-Anderage, England). Scanning electron microscopy (SEM) with energy dispersive X-ray analysis (EDX) was carried out using a TESCAN VEGA 3 SEM device (TESCAN, Edmonton, Alberta, Canada). Transmission electron microscopy (TEM) studies of the material obtained were performed using a JEM-2010 transmission electron microscope (University of Nebraska–Lincoln, Lincoln, NE, USA). To calculate Fe residual content in samples, the remaining mass, after thermogravimetry,

was multiplied by 0.7 in accordance with an approximation that all metallic Fe in samples converts to $Fe_2O_3$ during oxidation.

The marking of the samples' treatment and their treatment conditions are presented in Table 1.

**Table 1.** Marking of samples and conditions of their treatment.

| Sample | Description |
|--------|-------------|
| INI | Initial CNTs synthesis product |
| AT | INI after acid treatment |
| MO_0 | INI after mild oxidation without water at $T_{CNT}$ * = 400 °C and $t_{MO}$ ** = 6 h and acid treatment |
| MO_W1 | INI after mild oxidation with water at $T_{CNT}$ * = 400 °C, $T_{H2O}$ *** = 50 °C, $t_{MO}$ ** = 6 h and acid treatment |
| MO_W2 | INI after mild oxidation with water at $T_{CNT}$ * = 365 °C, $T_{H2O}$ *** = 50 °C, $t_{MO}$ ** = 6 h and acid treatment |
| MO_W3 | INI after mild oxidation with water at $T_{CNT}$ * = 400 °C, $T_{H2O}$ *** = 80 °C, $t_{MO}$ ** = 6 h and acid treatment |
| MO_W4 | INI after mild oxidation with water at $T_{CNT}$ * = 400 °C, $T_{H2O}$ *** = 80 °C, $t_{MO}$ ** = 12 h and acid treatment |

* $T_{CNT}$—temperature of mild oxidation, ** $t_{MO}$—time of mild oxidation, *** $T_{H2O}$—temperature of water bath during bubbling of purge gas.

## 3. Results

### 3.1. Free or Unbound Iron

To find out the quantity of free or unbound iron in CNTs synthesis products, they were treated by hydrochloric acid at 70 °C for 4 h, rinsed with distilled water until pH was higher 4 and then dried at 110 °C. It should be noted that 18 wt.% hydrochloric acid was used in these studies, while for chemical treatment of the same products, the authors in study [11] suggested to use concentrated 37 wt.% hydrochloric acid. Additional experiments showed that diluted 18 wt.% HCl is quite enough for chemical treatment of the samples in these studies and yields the same results as using concentrated 37 wt.% HCl. Thermogravimetry, Raman spectroscopy and scanning electron microscopy results of initial CNTs synthesis products are presented in Figures 1 and 2.

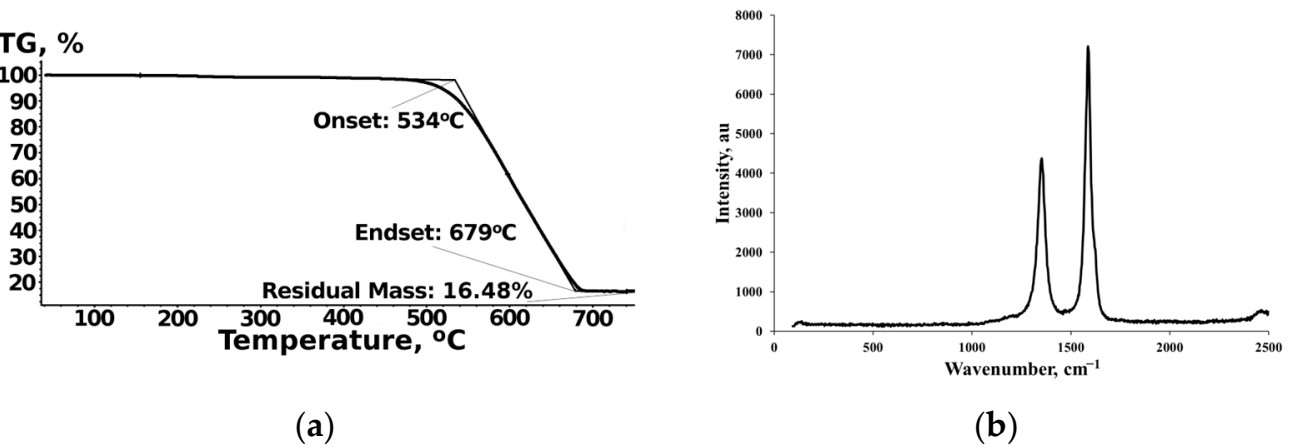

(**a**)  (**b**)

**Figure 1.** Thermogravimetry curve (**a**) and Raman spectrum (**b**) of initial CNTs synthesis products.

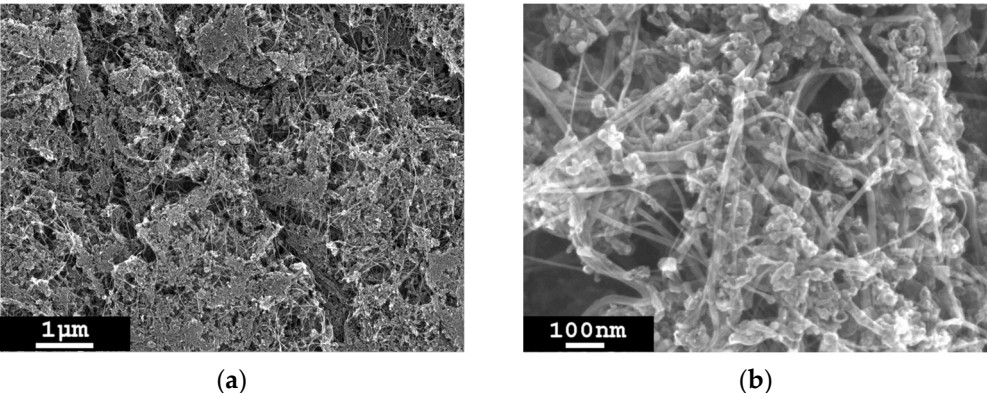

(**a**)  (**b**)

**Figure 2.** SEM images of initial CNTs synthesis product at 15,000 (**a**) and 100,000 (**b**) magnifications.

Thermogravimetry and Raman spectroscopy results of initial and acid treated samples are presented in Table 2.

**Table 2.** Thermogravimetry and Raman spectroscopy results of initial, acid treated samples and samples after mild oxidation with subsequent acid treatments.

| Sample | $T_{ONSET}$ [1], °C | $T_{ENDSET}$ [2], °C | m(Fe), wt.% | D/G [3] |
|--------|---------|---------|-------------|---------|
| INI | 534 | 679 | 11.5 | 0.66 |
| AT | 528 | 682 | 10.7 | 0.77 |
| MO_0 | 524 | 678 | 11.2 | 0.82 |
| MO_W1 | 523 | 684 | 8.1 [4] | 0.87 |
| MO_W2 | 531 | 703 | 9.1 [4] | 0.82 |
| MO_W3 | 550 | 726 | 3.8 | 0.95 |
| MO_W4 | 550 | 767 | 3.8 | 1.18 |

[1] point of intersection of interpolated virtual baseline and tangent drawn at point of inflection of near side of DTG peak. [2] point of intersection of interpolated virtual baseline and tangent drawn at point of inflection of far side of DTG peak. [3] ratio of D- and G-peak of Raman spectrum. [4] obtained from EDX data.

### 3.2. Water Influence at the Stage of Mild Oxidation

To determine the importance of water presence in purging gas during mild oxidation, two experiments were conducted with and without bubbling purging synthetic air through distilled water (MO_W1 and MO_0, respectively). The temperature of treatment was chosen as 400 °C to prevent undesired carbon oxidation by oxygen but to still be useable for water-assisted purification of SWCNTs (225–425 °C) [13]. Thermogravimetry and Raman spectroscopy results of acid treated and rinsed samples after mild oxidation at 400 °C with and without water are presented in Table 2.

### 3.3. Influence of Intensity of Mild Oxidation with Water

To find out the influence of mild oxidation intensity on the residual catalyst content, two experiments with different temperature of mild oxidation (365 °C and 400 °C, MO_W1 and MO_W2, respectively), different time of mild oxidation (6 and 12 h, MO_W3 and MO_W4, respectively) and different water saturation of purging gas (50 °C and 80 °C water bath temperature, MO_W1 and MO_W3, respectively) were conducted. Temperatures of mild oxidation were chosen in the investigated interval (225–425 °C) for SWCNTs [13]. The water content of the purge gas for the selected temperatures should increase by a factor of three when the temperature of the water bath changes from 50 to 80 °C. Thermogravimetry and Raman spectroscopy results of samples after mild oxidation with different degrees of water saturation of purging gas, mild oxidation time and temperature are presented in Table 2.

### 3.4. Oxidative Heat Treatment after Mild Oxidation

To investigate the influence of previous mild oxidation on the subsequent oxidation with oxygen, the samples were heat treated at 480 °C for 2 h under a static synthetic air atmosphere in a muffle furnace with subsequent acid treatment. Thermogravimetry and weight loss results of samples heat treated at 480 °C for 2 h are presented in Table 3.

**Table 3.** Thermogravimetry and weight losses results of samples heat treated at 480 °C for 2 h.

| Sample | $T_{ONSET}$, °C | $T_{ENDSET}$, °C | $\Delta m_{TO}$ *, % | m(Fe), wt.% |
|---|---|---|---|---|
| INI_HT | 547 | 715 | −34.5 | 4.6 |
| MO_0_HT | 542 | 674 | −30.9 | 3.1 |
| MO_W1_HT | 549 | 710 | −19.4 | 2.2 |
| MO_W2_HT | 547 | 731 | −20.7 | 2.0 |
| MO_W3_HT | 557 | 749 | −4.5 | 1.8 |
| MO_W4_HT | 550 | 746 | −4.6 | 1.9 |

* Weight losses after heat treatment at 480 °C for 2 h and acid treatment.

### 3.5. Removal of Non-CNT Particles from the Sample

To burn out all non-CNTs particles form CNTs synthesis products, the samples rinsed and dried after mild oxidation with water were heat treated under static air atmosphere at 480 °C [12] for 6 h, then acid treated and dried. Raman spectroscopy and scanning electron microscopy results of acid treated and rinsed samples after mild oxidation, both before and after heat treatment at 480 °C for 6 h, are presented in Figures 3 and 4.

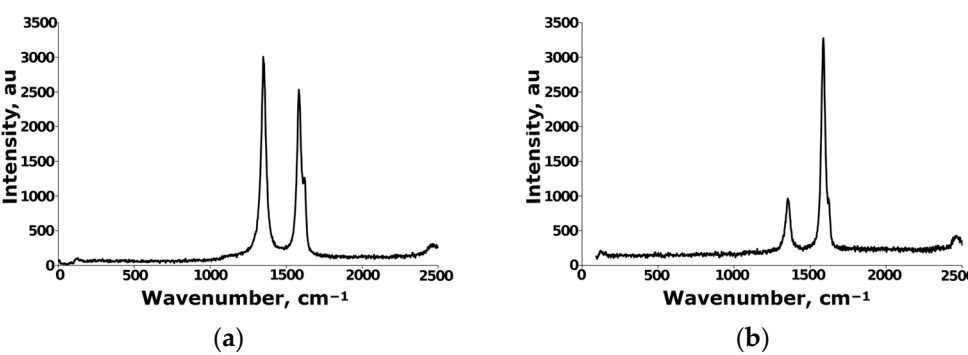

(a)                    (b)

**Figure 3.** Raman spectra of acid treated and rinsed samples after mild oxidation before (**a**) and after (**b**) heat treatment at 480 °C for 6 h.

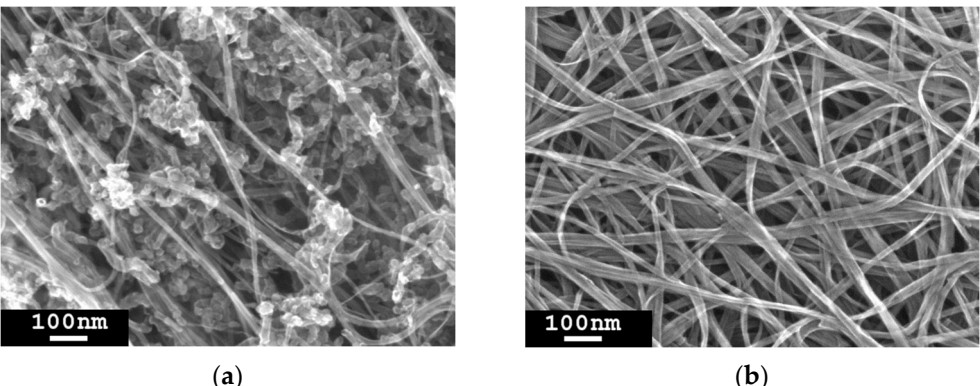

(a)                    (b)

**Figure 4.** SEM images of acid treated and rinsed samples after mild oxidation before (**a**) and after (**b**) heat treatment at 480 °C for 6 h.

## 4. Discussion

As one can see from Table 2, the hydrochloric acid treatment of initial synthesis products leads to an insignificant decrease of iron content in the sample (see INI and AT samples). On the other hand, according to scanning electron microscopy results, acid treatment does not cause significant structural changes, but, in accordance with Table 2, results in slight increase of D/G ratio of Raman spectra. It is known [16] that Raman spectroscopy is a powerful method for the investigation of nanostructured carbon materials, including CNTs, and that it can be used to roughly estimate UCNTs purity from the D/G ratio of Raman spectra [17], where the G band is related to absorbance at ~1584 cm$^{-1}$ and the D band—at 1300–1400 cm$^{-1}$. For example, the increasing of the D/G ratio of Raman spectra can be related to the increasing of edge defects of nanotubes [17], which can be created after partial removal of catalyst particles.

All mentioned findings likely show that only a small amount of residual iron catalyst is unbound and accessible to acid, but the main part of the residual iron catalyst is encapsulated inside CNTs or non-CNTs particles in the form of $\alpha$-Fe or Fe$_3$C particles [18] and cannot be removed from the sample by simple acid treatment without additional high temperature treatment. It can be proposed that mild oxidation by wet air can be used to remove iron catalyst particles from UCNTs as suggested by the successful application of similar techniques for SWCNTs [13]. The validity of this assumption and importance of water during mild oxidation for UCNTs synthesis products can be seen in Table 2, where the results of two experiments carried out at 400 °C with (MO_1 sample) and without (MO_0 sample) water in purging oxidative gas are presented.

As can be seen from Table 2, the presence of water in purging oxidative gas causes a removal of more iron catalyst particles from the sample after sequential acid treatment when compared to the sample without water in purging oxidative gas (see results for MO_1 and MO_0 samples, respectively). On the other hand, the presence of water in purging oxidative gas does not result in significant changes in sample structure but causes the increasing of the D/G ratio of Raman spectra (Table 2). The increasing of the D/G ratio due to water presence in purging gas can be related to the appearance of more CNT edge defects [17] after catalyst removal as well as a result of inner structural strain formation inside the carbon matrix [19]. It is known that during oxidation of iron by oxygen, which can be expressed by a schematic reaction route (1) [20], the density is changed by about 35% when iron transforms from a metallic state to iron (III) oxide [12], and that this change can result in inner structural strain formation in cases where iron is located inside carbon shells, such as inside carbon nanotubes.

It was shown earlier [16] that iron catalyst particles inside CNTs synthesis products obtained by CCVD methods under similar experimental conditions can be in the form of metallic iron or iron carbides, namely, cementite Fe$_3$C [18]. The type of iron that exists inside CNTs as Fe$_3$C ($\rho$ = 7.68 g/cm$^3$ [21]) will be transformed during oxidation to Fe$_2$O$_3$ with practically the same density changes and chemical reaction, which can be expressed by schematic reaction route (2) [22].

$$\text{Fe}_{(S)} \rightarrow \text{FeO}_{(S)} \rightarrow \text{Fe}_3\text{O}_{4(S)} \rightarrow \text{Fe}_2\text{O}_{3(S)} \tag{1}$$

$$\text{Fe}_3\text{C}_{(S)} \rightarrow \text{Fe}_x\text{O}_{y(S)} + \text{CO}, \text{CO}_{2(G)} \tag{2}$$

It should be noted that mild oxidation with water results in sufficient increase of oxygen content in the samples, as can be seen from EDX data (Table 4), while oxidation without water causes only a slight increase of oxygen content.

**Table 4.** Fe/O and C/O molar ratio according to EDX analysis.

| Sample | C/O Molar Ratio | Fe/O Molar Ratio |
|--------|-----------------|------------------|
| INI | 87.8 | 3.1 |
| MO_0 | 72.6 | 2.8 |
| MO_W1 | 19.0 | 0.7 |
| MO_W3 | 15.9 | 0.4 |
| MO_W4 | 13.2 | 0.3 |

Moreover, the formation of $Fe_2O_3$ from Fe inside CNT cups during mild oxidation with water can be seen from TEM images (Figure 5). The FFT pattern of initial CNT synthesis products is shown in the inset of Figure 5a, which was indexed as a base-centered cubic crystal structure. The analysis showed that it corresponds to $\alpha$-Fe. The FFT pattern of CNT synthesis products after mild oxidation with water is shown in the inset of Figure 5b, which was indexed as a cubic crystal structure. The analysis showed that it corresponds to $Fe_2O_3$.

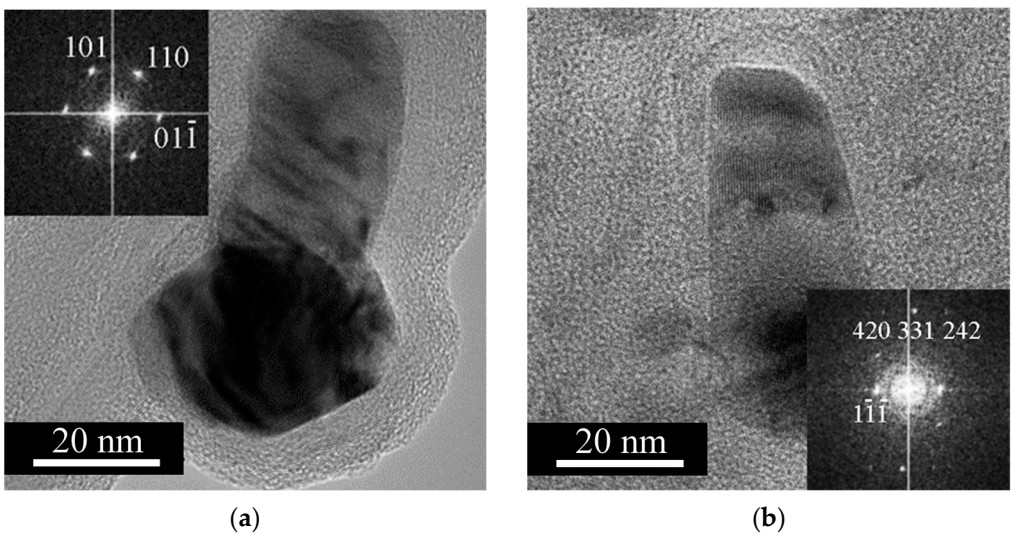

(**a**)　　　　　　　　　　　　　　　　　　(**b**)

**Figure 5.** TEM images of CNT end cup with encapsulated catalyst particles before (**a**) and after (**b**) mild oxidation with water and FFT images of these particles proving $\alpha$-Fe (**a**) and $Fe_2O_3$ (**b**).

It should be noted that the presence of water can be a reason for the formation of different oxide hydroxides, as one can see from the schematic reaction route (3). Moreover, according to study [23], water can change the kinetics of iron oxidation by air and change the distribution of iron oxides (hematite $Fe_2O_3$, magnetite $Fe_3O_4$, etc.) inside oxidation products.

$$Fe_{(S)} + (x + y)/2O_{2(G)} + y/2H_2O_{(G)} \rightarrow FeO_x(OH)_{y(S)} \tag{3}$$

It should be stressed that even though the Fe/O molar ratio, obtained from EDX data (Table 4) and corresponding to $FeO_{2.8}$ composition, suggests the existence of some part of iron oxide hydroxides or even iron hydroxides inside the samples after mild oxidation with water, iron hydroxides and iron oxide hydroxides should be decomposed at 400 °C [24]. Considering this, one can suggest that a high oxygen content may be related to the formation of oxygen-containing surface functional groups on CNT and non-CNT particles. For example, according to XPS studies [25], the heat treatment with air of the same CNTs synthesis products can result in formation of additional hydroxyl (-OH), carbonyl (-C=O), carboxyl (-COOH), ester (-COOC-) and ether (-COC-) groups.

In case where iron is not oxidized completely and there are many carbon shells enveloping these iron particles, the structural strain formed during iron oxidation does not

cause the destruction of carbon shells, and so prevents encapsulated iron or iron carbides particles being removed by subsequent acid treatment. It can be concluded that the depth of iron oxidation can be dependent on mild oxidation conditions, such as temperature and time of mild oxidation, as well as the degree of purging oxidative gas saturation by water molecules. To show this, three additional experiments with different intensities of mild oxidation were carried out (MO_W2, MO_W3 and MO_W4).

As can be seen in Table 2, the increase of mild oxidation intensity (at water bath temperature, $T_{H2O}$ = 50 °C) by increasing the mild oxidation temperature from 365 to 400 °C (MO_W2 and MO_W1) causes a decrease in residual iron content in the samples from 9.1 to 8.1 wt.% and an enhancement of the D/G ratio of Raman spectra from 0.82 to 0.87. Enhancing the degree of purging gas saturation by water (MO_W1 and MO_W3) by increasing water bath temperature from 50 to 80 °C (at mild oxidation temperature, $T_{MO}$ = 400 °C) results in a decrease in residual iron content in the samples from 8.1 to 3.8 wt.% and an enhancement of the D/G ratio of Raman spectra from 0.87 to 0.95. On the other hand, increasing the mild oxidation time at $T_{MO}$ = 400 °C and $T_{H2O}$ = 80 °C (see samples MO_W3 and MO_W4) does not cause additional iron removal, but instead results in additional enhancement of the D/G ratio from 0.95 to 1.18 (Table 2) and oxygen content from C/O = 15.9 to 13.2 (Table 4). So, one can suppose that intensifying mild oxidation, up to some degree, results in improving iron catalyst particles removal from those samples that can be related with more complete oxidation of iron and, consequently, more complete destructions of carbon shells, which envelop iron catalyst particles. On the other hand, too intensive mild oxidation does not cause additional iron removal, but instead results in an increase of carbon matrix defectiveness by formation of oxygen-containing surface functional groups.

It should be noted that mild oxidation with wet air allows the removal of iron catalyst particles without significant losses of different forms of carbon due to carbon oxidation. For example, as one can see in Table 5, heat treatment with dry air (see HT_480-2 and HT_480-6 samples) allows the same level of residual iron content to be reached with more than twice the weight loss when compared with mild oxidation using wet air (see MO_400-6 sample).

**Table 5.** Weight losses during purification using wet and dry air with subsequent acid treatment.

| Sample | m(Fe), wt.% | Δm, % |
|---|---|---|
| MO_400-6 * | 3.8 | −12.9 |
| HT_480-2 ** | 4.6 | −34.5 |
| HT_480-6 *** | 3.1 | −83.0 |

* Mild oxidation with wet air at 400 °C for 6 h, ** Heat treatment with dry air at 480 °C for 2 h, *** Heat treatment with dry air at 480 °C for 6 h.

In other words, lower level of iron content before heat treatment is undertaken is necessary to increase the yield of CNTs, because during heat treatment iron particles catalyze carbon oxidation [3] and results in the oxidation of useful CNTs. For example, as can be seen from Figure 6, the weight loss during heat treatment under static air atmosphere at 480 °C for 2 h is proportional to residual iron content in the samples of CNTs synthesis products unrelated to previous treatment.

It should be noted that wet air mild oxidation (MO_W3_HT) results in a reduction of residual iron content by more than 140% when compared to the sample without mild oxidation (INI_HT), and by more than 63% when compared to the sample with dry air mild oxidation (MO_0_HT). Moreover, the more intensive the conditions of wet air mild oxidation (MO_W2 < MO_W1 < MO_W3), the less residual iron in the sample (Table 3).

Additional high temperature treatment with sequential acid treatment of samples after mild oxidation and acid treatment allows us to obtain ultralong CNTs without significant content of non-CNTs and with a residual iron content as low as 0.9 wt.% and a D/G ratio as low as 0.26 (Figures 3 and 4).

The results obtained allow us to conclude that mild oxidation with wet air at the initial stage of purification makes it possible to remove the majority of iron catalyst particles without significant carbon losses while increasing the yield of CNTs after heat treatment for non-CNTs removal.

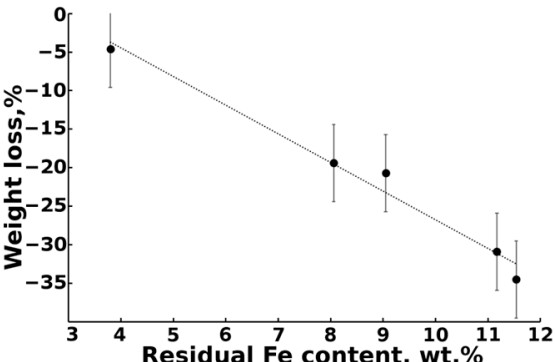

**Figure 6.** Dependence of weight losses during heat treatment of samples under static air atmosphere on residual iron content.

## 5. Conclusions

The importance of water during purification of ultralong CNTs from iron catalyst particles obtained by catalytic CVD method is shown. The presence of water in the purging oxidative gas during mild oxidation at 400 °C allows the oxidation of most encapsulated iron particles without significant oxidation of different forms of carbon. The oxidation of encapsulated iron particles causes density to decrease and inner structural strain formation that results in destruction of carbon shells enveloping iron particles, allowing them to be removed after subsequent acid treatment. Factors like temperature and degree of saturation of purging oxidative gas with water vapor can be used to intensify the process of iron removal. On the other hand, too prolonged mild oxidation can cause formation of oxygen containing functional groups on the surfaces of CNTs and non-CNT particles and results in defect formation on carbon matrix but does not lead to the removal of additional catalyst particles from the samples.

The purification of CNTs synthesis products under mild oxidation with the assistance of water vapor can be used to develop economic and ecologically friendly methods for purification of ultralong carbon nanotubes obtained by catalytic chemical vapor deposition with iron catalyst.

**Author Contributions:** Conceptualization, A.R.K. and M.A.K.; methodology, A.R.K. and M.A.K.; validation, V.Z.M.; investigation, M.A.K., V.V.D., B.A.K. and V.A.N.; writing—original draft preparation, M.A.K.; writing—review and editing, V.Z.M.; supervision, V.Z.M. and A.R.K.; project administration, V.Z.M. All authors have read and agreed to the published version of the manuscript.

**Funding:** The work was carried out using the equipment of FSBI TISNCM SUEC "Structural Measurements in the Laboratory of the Department of Structural Research". This work was supported through State Assignment # FNRW-2022-0002.

**Data Availability Statement:** The data presented in this article are available on request from the corresponding authors.

**Acknowledgments:** The authors would like to acknowledge Nikita V. Kazennov for samples supplying, Sergey A. Urvanov for discussion, Taisia E. Drozdova for Raman measurements, Natalia I. Batova for SEM/EDX measurements.

**Conflicts of Interest:** The authors declare no conflict of interest.

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
