# Peer review of "The Importance of Water for Purification of Longer Carbon Nanotubes for Nanocomposite Applications"

_jcs, doi:10.3390/jcs7020079_

Round 1

Reviewer 1 Report

In this paper, the authors studied the influence of water vapor existence in wet air under mild conditions, excluding carbon oxidation at 225-425°C, followed by acid treatment for purification of ultralong carbon nanotubes (UCNTs). With the help of water vapor, the residual iron catalyst content and non-CNT particle amount decreased significantly without impacting the yield of the desired product, UCNTs. The authors claimed that decreasing catalyst and non-CNT particles resulting from oxidation causes swelling of encapsulated iron, destruction of carbon shell, and exposure of iron for the following acid treatment.

But there are some comments the authors needed to claim:

(1)   The authors used Fe's residual content to represent the catalyst's residual content. When the content of Fe is highly decreased with the help of water vapor, the authors conclude that the water vapor will decrease the residual catalyst content. However, they are different types of catalysts used as UCNTs catalyst in chemical vapor deposition (CCVD) synthesis methods. The improvement of the yield of UCNTs with the help of water vapor in Fe catalyst cannot stand for other catalysts. The authors can only conclude that the existence of water vapor in mild conditions can help remove the Fe catalyst, not other catalysts.

(2)   In Page 1 Line 18, the authors speculated that “iron catalyst particles transform into iron oxides and hydroxides” after wet air treatment. However, the authors didn’t provide any evidence in the manuscript to prove that Fe had converted to iron oxides and iron hydroxides. To prove this claim, for example, the authors can provide the X-ray diffraction (XRD) data of initial CNTs synthesis product (INI), and INI after mild oxidation with water product. If iron peaks existed in INI sample, iron oxides and hydroxides peaks existed in INI after mild oxidation with water product, and then the authors can firmly conclude that “iron catalyst particles transform into iron oxides and hydroxides” after wet air treatment.

(3)   After mild oxidation with water product, how do Fe catalyst particles transform into iron oxides and hydroxides? Can the authors provide the chemical reaction equations? Reference [10] supports the formation of iron oxides. Is there any references to support the formation of iron hydroxides? Can authors provide any evidence to prove the formation of iron hydroxides?

(4)   In Page 2 Line 47, the authors claim that “for example, non-CNTs are easily oxidized at 480 - 500°C, while single wall CNTs (SWCNTs) begin to oxidize at higher temperatures.” It is better to put the specific oxidization temperatures of SWCNTs here. Readers have no idea how high a “higher temperature” is. Meanwhile, the authors used 480°C temperature in the heat treatment process to burn out all non-CNTs particles. If non-CNTs are easily oxidized at 480 - 500°C, why did the authors use 480°C to burn out non-CNTs, not 500 °C?

(5)   In Page 2 Table 1, the authors used two temperatures during mild oxidation, 400 °C, and 365 °C. Can authors briefly explain why they chose these two temperatures? Meanwhile, in Page 3 Table 2, why does the MO_W2 sample has a lower Fe content than the MO_W1 sample? In other words, why can 365 °C in mild oxidation decrease more iron content than 400 °C in mild oxidation? In Page 5 Line 146, the authors claim that “the increasing of D/G ratio of Raman spectra can be related to the increasing of edge defects of nanotubes which can be created after partial removal of catalyst particles.” Then why does the MO_W2 sample has a lower D/G ratio than the MO_W1 sample when the MO_W2 sample has a lower Fe content?

(6)   In the Discussion section, the authors need to specify the samples’ names when comparing different samples and draw a conclusion. For example, section 3.2, “To find out the importance of water presence in purging gas during mild oxidation, two experiments were carried out.” The authors need to specify that they operate MO_0 and MO_W1 here to determine water’s importance. Especially Page 5 Line 139, to emphasize the importance of hydrochloric acid treatment, Page 5 Line 156 for water, Page 6 Line 176 for mild oxidation, and Page 6 Line 206 for wet air mild oxidation.

(7)   In Page 4 Section 3.3, the authors focused on the influence of the intensity of mild oxidation with water. MO_W1 and MO_W1 samples can be used to compare the influence of different temperatures of mild oxidation (365°C and 400°C). Which samples can be used to compare the influence of different times of mild oxidation (6 and 12 hours) or different water saturation of purging gas, respectively? For MO_W1 and MO_W3, the time of mild oxidation and water saturation is different. We need to control the variables to conclude. The authors can design a new sample, MO_W4, INI, after mild oxidation with water at TCNT*=400 °C, TH2O***=80°C, tMO**=6 hours, and acid treatment. MO_W1 and MO_W4 can tell the influence of water saturation, and MO_W3 and MO_W4 can tell the influence of time of mild oxidation.

(8)   In Page 4, Section 3.4 and 3.5 share the same title. Based on the content of Section 3.5, if the authors want to conclude the influence of heat treatment, scanning electron microscopy images of the sample with and without heat treatment should be provided.

(9)   In Page 5 Line 150, the authors claimed that the “main part of the residual iron catalyst is encapsulated inside CNTs or non-CNTs particles and cannot be removed from the sample by simple acid treatment without additional high-temperature treatment.” Although Table 2 provided the content of Fe before and after acid treatment, it does not indicate the phase of Fe sources. XRD data of the samples before and after acid treatment are needed to prove the authors’ claim.

Author Response

Dear Reviewer,

Thank you very much for attentive reading of the article and expressing very useful remarks. We have revised the article in accordance with your comments and would like to answer all your questions. 

Question (1). The authors used Fe's residual content to represent the catalyst's residual content. When the content of Fe is highly decreased with the help of water vapor, the authors conclude that the water vapor will decrease the residual catalyst content. However, they are different types of catalysts used as UCNTs catalyst in chemical vapor deposition (CCVD) synthesis methods. The improvement of the yield of UCNTs with the help of water vapor in Fe catalyst cannot stand for other catalysts. The authors can only conclude that the existence of water vapor in mild conditions can help remove the Fe catalyst, not other catalysts.

Response (1). The authors have stressed in the abstract and in the main body that the article dealing with only iron catalyst.

Question (2).   In Page 1 Line 18, the authors speculated that “iron catalyst particles transform into iron oxides and hydroxides” after wet air treatment. However, the authors didn’t provide any evidence in the manuscript to prove that Fe had converted to iron oxides and iron hydroxides. To prove this claim, for example, the authors can provide the X-ray diffraction (XRD) data of initial CNTs synthesis product (INI), and INI after mild oxidation with water product. If iron peaks existed in INI sample, iron oxides and hydroxides peaks existed in INI after mild oxidation with water product, and then the authors can firmly conclude that “iron catalyst particles transform into iron oxides and hydroxides” after wet air treatment.

Response (2). The authors have added results of additional EDX and FFT experiments, which proves the transformation of Fe to Fe2O3 during mild oxidation with water. The hypothesis of hydroxides formation is based on high Fe/O molar ratio, but in the article, the authors have added remarks that all hydroxides and oxide hydroxides should be decomposed at temperature of mild oxidation (400°C). So the authors think that the oxide formation is proven, while hydroxide formation is probable.

Question (3).  After mild oxidation with water product, how do Fe catalyst particles transform into iron oxides and hydroxides? Can the authors provide the chemical reaction equations? Reference [10] supports the formation of iron oxides. Is there any references to support the formation of iron hydroxides? Can authors provide any evidence to prove the formation of iron hydroxides?

Response (3). Possible chemical reaction equations have been added to the article. The discussion about possible iron hydroxides formation has been added to the article.

Question (4).   In Page 2 Line 47, the authors claim that “for example, non-CNTs are easily oxidized at 480 - 500°C, while single wall CNTs (SWCNTs) begin to oxidize at higher temperatures.” It is better to put the specific oxidization temperatures of SWCNTs here. Readers have no idea how high a “higher temperature” is. Meanwhile, the authors used 480°C temperature in the heat treatment process to burn out all non-CNTs particles. If non-CNTs are easily oxidized at 480 - 500°C, why did the authors use 480°C to burn out non-CNTs, not 500 °C?

Response (3). Since the authors of [10 or R1] do not specify the temperature of SWCNT oxidation, we prefer to withdraw this part of the statement from the manuscript. Concerning the Reviewer’s question on alternative 480 vs 500°C, we should draw your attention to the presence of Fe residual catalysts, which accelerates oxidation and forces us to be as close to the bottom of the possible temperature range as possible.   In addition to that, according to literature [R2], non-CNTs are easily oxidized at 480-500°C without presence of catalyst.

Question (5). In Page 2 Table 1, the authors used two temperatures during mild oxidation, 400 °C, and 365 °C. Can authors briefly explain why they chose these two temperatures? Meanwhile, in Page 3 Table 2, why does the MO_W2 sample has a lower Fe content than the MO_W1 sample? In other words, why can 365 °C in mild oxidation decrease more iron content than 400 °C in mild oxidation? In Page 5 Line 146, the authors claim that “the increasing of D/G ratio of Raman spectra can be related to the increasing of edge defects of nanotubes which can be created after partial removal of catalyst particles.” Then why does the MO_W2 sample has a lower D/G ratio than the MO_W1 sample when the MO_W2 sample has a lower Fe content?

Response (5). The choice of temperatures was based on the article for SWCNTs [R1], but our own experiments showed small iron content changes at 365°C (from 11.5 to 9.1 wt.%Fe) when MWCNTs were used. Taking this into account the temperature was increased up to 400°C. The authors have checked results and found out that there were quite big mistake for MO_W2 and MO_W3 because the results were obtained from thermogravimetry and small masses were used. Taking this into account the authors have measured iron content in these samples by EDX methods.

Question (6).   In the Discussion section, the authors need to specify the samples’ names when comparing different samples and draw a conclusion. For example, section 3.2, “To find out the importance of water presence in purging gas during mild oxidation, two experiments were carried out.” The authors need to specify that they operate MO_0 and MO_W1 here to determine water’s importance. Especially Page 5 Line 139, to emphasize the importance of hydrochloric acid treatment, Page 5 Line 156 for water, Page 6 Line 176 for mild oxidation, and Page 6 Line 206 for wet air mild oxidation.

Response (6). The demanded information has been added to the article.

Question (7).   In Page 4 Section 3.3, the authors focused on the influence of the intensity of mild oxidation with water. MO_W1 and MO_W1 samples can be used to compare the influence of different temperatures of mild oxidation (365°C and 400°C). Which samples can be used to compare the influence of different times of mild oxidation (6 and 12 hours) or different water saturation of purging gas, respectively? For MO_W1 and MO_W3, the time of mild oxidation and water saturation is different. We need to control the variables to conclude. The authors can design a new sample, MO_W4, INI, after mild oxidation with water at TCNT*=400 °C, TH2O***=80°C, tMO**=6 hours, and acid treatment. MO_W1 and MO_W4 can tell the influence of water saturation, and MO_W3 and MO_W4 can tell the influence of time of mild oxidation.

Response (7). The authors have carried out additional experiments at TCNT=400 °C, TH2O=80°C, tMO=6 hours and named it as MO_W3. The previous MO_W3 that were carried out at  TCNT=400 °C, TH2O=80°C, tMO=12 hours were renamed to MO_W4. All results have been added and conclusions have been made.

Question (8). In Page 4, Section 3.4 and 3.5 share the same title. Based on the content of Section 3.5, if the authors want to conclude the influence of heat treatment, scanning electron microscopy images of the sample with and without heat treatment should be provided.

Response (8). Title of 3.5 has been changed, but additional SEM images have been provided to the main body of article.

Question (9).   In Page 5 Line 150, the authors claimed that the “main part of the residual iron catalyst is encapsulated inside CNTs or non-CNTs particles and cannot be removed from the sample by simple acid treatment without additional high-temperature treatment.” Although Table 2 provided the content of Fe before and after acid treatment, it does not indicate the phase of Fe sources. XRD data of the samples before and after acid treatment are needed to prove the authors’ claim.

Response (9). The phase of Fe sources inside CNT and carbon particles was thoroughly investigated in our previous research and published elsewhere [R3]. This information is added to the main body of article.

[R1] Chiang, I.W.; Brinson, B.E.; Huang, A.Y.; Willis, P.A.; Bronikowski, M.J.; Margrave, J.L.; Smalley, R.E.; Hauge, R.H. Purifica-tion and Characterization of Single-Wall Carbon Nanotubes (SWNTs) Obtained from the Gas-Phase Decomposition of CO (HiPco Process). J. Phys. Chem. B 2001, 105, 8297-830. https://doi.org/10.1021/jp0114891

[R2] Rouf, S.A.; Usman, Z.; Masood, H.T.; Majeed, A.M.; Sarwar, M.; Abbas, W. Synthesis and Purifiation of Carbon Nanotubes. In Book Carbon Nanotubes - Redefining the World of Electronics, 2nd ed.; Ghosh, P.K., Datta, K., Rushi, A.D., Eds.; Publisher: IntechOpen London, The United Kingdom, 2022; Volume 3, pp. 415–479. https://doi.org/10.5772/intechopen.98221

[R3] Kulnitskiy, B.; Karaeva, A.; Mordkovich, V.; Urvanov, S.; Bredikhina A. TEM studies of conical scroll carbon nanotubes formed by aerosol synthesis. IOP Conf. Series: Materials Science and Engineering 2019, 693, 012017. https://doi.org/10.1088/1757-899X/693/1/012017

Reviewer 2 Report

The current manuscript by Mordkovich et al. has reported the role of water in purification of ultralong carbon nanotubes, which could be utilized to improve future nanocomposite quality. Their research is very practical and could be a helpful reference for future studies. Also, the scope of this work fits the journal well. Below I have attached my suggestions and I highly recommend that the authors could revise accordingly:

Major

1. Section 3 Result (page 3-4, line 87-132): Most statements in this section is a brief introduction for each experiment condition setup and statements leading to figures and tables. It is highly recommended that the authors can further describe the results and give some brief explanation or discussion that can lead to a conclusion. The readers may not only want to see the results but also want to see how the authors explain their results and what kind of conclusion can be retrieved from these numbers. This kind of reasoning is recommended for all subsections (3.1-3.5) in result section.

2. Page 4, line 109-111: MO_W1 and MO_W3 may not be convincing enough in studying the impact of oxidation time and water saturation individually since the authors have changed two conditions, time of mild oxidation and water bath temperature together, which is not a good variable control. It is suggested to add one more group with conditions such as TCNT=400C, TH2O=80C, tMO=6 hours.

3. Data is missing in table 3. There is no data related to weight loss after mild oxidation and acid treatment exhibited.

4. Subsections 3.4 & 3.5, page 4: Subsections 3.4 and 3.5 are all discussing the influence of oxidative heat treatment. The only difference is the time of treatment. The authors may want to combine them together and re-write this subsection. Since the authors mentioned the time of oxidative heat treatment could be 2 or 6 hours, the authors may also want to compare the weight loss between them too.

5. Additional figures are recommended: The authors only show SEM images, thermogravimetry results, and Raman spectroscopy results for initial conditions and heat treatment after mild oxidation conditions. Therefore, it is hard to provide readers with a straightforward view regarding impacts of multiple factors that the authors have discussed. It is highly recommended that the authors could provide SEM images, thermogravimetry curves, and Raman spectra of other conditions that have been discussed in this article or in their supplemental materials.

Minor

1. Current figures need improvement: notations in all figures are very small and could be difficult to read. The authors may want to re-produce their figures and also enlarge them if needed. These notations include numbers and words for the axes, for the special remarks in the figure, etc.

2. Page 2, line 51-52: It seems that the authors have something missing here, “According to [11] SWCNTs can…”. I expect there should be a name or something else ahead of “[11]”.

3. Table 2: The authors may want to provide a brief explanation of TONSET, TENDSET, D/G, so that readers who are not familiar with these parameters can know the concept behind these abbreviations in parameters.

4. Section 4 Discussion, line 141, page 5: Figure 1 is regarding initial CNT and there are no SEM images for acid treatment condition. The authors need to provide additional SEM images or other evidence to support this conclusion. Addressing comment #5 in major could help to solve it.

Author Response

Dear Reviewer,

Thank you very much for attentive reading of the article and expressing very useful remarks. We have revised the article in accordance with your comments and would like to answer all your questions. 

Major

Question 1. Section 3 Result (page 3-4, line 87-132): Most statements in this section is a brief introduction for each experiment condition setup and statements leading to figures and tables. It is highly recommended that the authors can further describe the results and give some brief explanation or discussion that can lead to a conclusion. The readers may not only want to see the results but also want to see how the authors explain their results and what kind of conclusion can be retrieved from these numbers. This kind of reasoning is recommended for all subsections (3.1-3.5) in result section.

Response 1. The authors have expanded the main body and tried to explain the results in more detail.

Question 2. Page 4, line 109-111: MO_W1 and MO_W3 may not be convincing enough in studying the impact of oxidation time and water saturation individually since the authors have changed two conditions, time of mild oxidation and water bath temperature together, which is not a good variable control. It is suggested to add one more group with conditions such as TCNT=400C, TH2O=80C, tMO=6 hours.

Response 2. The authors have carried out additional experiment at TCNT=400°C, TH2O=80°C, tMO=6 hours and named it MO_W3. The previous MO_W3 that were carried out at  TCNT=400°C, TH2O=80°C, tMO=12 hours were renamed to MO_W4. All results have been added and conclusions have been made 

Question 3.Data is missing in table 3. There is no data related to weight loss after mild oxidation and acid treatment exhibited.

Response 3. The table 3 was edited in accordance to this comment.

Question 4. Subsections 3.4 & 3.5, page 4: Subsections 3.4 and 3.5 are all discussing the influence of oxidative heat treatment. The only difference is the time of treatment. The authors may want to combine them together and re-write this subsection. Since the authors mentioned the time of oxidative heat treatment could be 2 or 6 hours, the authors may also want to compare the weight loss between them too.

Response 4. The title of 3.5 section has been changed with providing additional information.

Question 5. Additional figures are recommended: The authors only show SEM images, thermogravimetry results, and Raman spectroscopy results for initial conditions and heat treatment after mild oxidation conditions. Therefore, it is hard to provide readers with a straightforward view regarding impacts of multiple factors that the authors have discussed. It is highly recommended that the authors could provide SEM images, thermogravimetry curves, and Raman spectra of other conditions that have been discussed in this article or in their supplemental materials.

Response 5. The data on all conditions under discussion are summarized in Tables 2 to 5. Illustration of these results by figures/pictures cannot be as representative as tables due to no visual difference between SEM/Raman/TG pictures in step-by-step representation. We believe it important to show the comparison of initial and final conditions, where it is clearly visible. With that purpose we re-edited our picture representation and introduced new images into Figure 3, Figure 4 and introduced TEM data as Figure 5. We think it answers the concerns by the Reviewer.

Minor

Question 1. Current figures need improvement: notations in all figures are very small and could be difficult to read. The authors may want to re-produce their figures and also enlarge them if needed. These notations include numbers and words for the axes, for the special remarks in the figure, etc.

Response 1. The figures have been improved.

Question 2. Page 2, line 51-52: It seems that the authors have something missing here, “According to [11] SWCNTs can…”. I expect there should be a name or something else ahead of “[11]”.

Response 2. The authors have changed to “According to study [11], SWCNTs can…”

Question 3. Table 2: The authors may want to provide a brief explanation of TONSET, TENDSET, D/G, so that readers who are not familiar with these parameters can know the concept behind these abbreviations in parameters.

Response 3. Additional information has been added.

Question 4. Section 4 Discussion, line 141, page 5: Figure 1 is regarding initial CNT and there are no SEM images for acid treatment condition. The authors need to provide additional SEM images or other evidence to support this conclusion. Addressing comment #5 in major could help to solve it.

Response 4.  This question was addressed in the response to comment #5 of the major list

Round 2

Reviewer 1 Report

1. Please increase the font size of Figure 1, 3, 5.

Author Response

Dear Reviewer,

Thank you very much once again for spending your precious time for attentively reading of our article and your useful comments. Please find answers to your comments as follows:

Question 1: Please increase the font size of Figure 1, 3, 5.

Respond 1: The font size of Figures 1, 3 and 6 have been increased.

Reviewer 2 Report

The authors have addressed the majority of my previous comments. However, some additional minor revisions are needed:

1.  Table 3: there is no data shown in this table related to the notation below " * Weight losses after mild oxidation and acid treatment".

2.  Page 7 line 183 and table 4: numbers here (density of Fe3C and data in the table) may have the misusage between comma "," and dot ".". Please double check if there are any typos here.

3. Page 8 line 210: there might be an additional "that" in the sentence as a typo.

4. Table 5: one of the notations below the table is not revised accordingly, " * mild oxidation with wet air at 400°C for 12 hours".

Author Response

Dear Reviewer,

Thank you very much once again for spending your precious time for attentively reading of our article and your useful comments. Please find answers to your comments as follows:

Question 1: Table 3: there is no data shown in this table related to the notation below " * Weight losses after mild oxidation and acid treatment".

Respond 1: The correction in accordance with Question 1 has been done in the manuscript, namely, the phrase “Weight losses after mild oxidation and acid treatment” has been deleted from text.

Question 2: Page 7 line 183 and table 4: numbers here (density of Fe3C and data in the table) may have the misusage between comma "," and dot ".". Please double check if there are any typos here.

Respond 2: All decimal separators have been changed to the dots in the manuscript. 

Question 3: Page 8 line 210: there might be an additional "that" in the sentence as a typo.

Respond 3: The double “that” has been deleted from the manuscript.

Question 4: Table 5: one of the notations below the table is not revised accordingly, " * mild oxidation with wet air at 400°C for 12 hours".

Respond 4: The notation below Table 5 has been changed from 12 hours to 6 hours.

Round 3

Reviewer 1 Report

1. In Page 4 Table 2, some D/G values have three decimal places but some have two. Please keep it uniform.

2. In Page 5 Table 3, why 1.988 (the value of m(Fe) of MO_W4_HT) is rounded to 1.9, rather than 2?

Reviewer 2 Report

The authors have addressed all my comments and I have no further feedback. Thanks for authors' great patience in revision and it is my pleasure to review this paper.